# Setd2 inactivation sensitizes lung adenocarcinoma to inhibitors of oxidative respiration and mTORC1 signaling

David M. Walter[1,2,5,6], Amy C. Gladstein [1,2,6], Katherine R. Doerig[1,2], Ramakrishnan Natesan[1], Saravana G. Baskaran[1], A. Andrea Gudiel[1], Keren M. Adler[1,2], Jonuelle O. Acosta[1,2], Douglas C. Wallace[3], Irfan A. Asangani[1,2,4] & David M. Feldser [1,2,4✉]

*SETD2* is a tumor suppressor that is frequently inactivated in several cancer types. The mechanisms through which SETD2 inactivation promotes cancer are unclear, and whether targetable vulnerabilities exist in these tumors is unknown. Here we identify heightened mTORC1-associated gene expression programs and functionally higher levels of oxidative metabolism and protein synthesis as prominent consequences of *Setd2* inactivation in KRAS-driven mouse models of lung adenocarcinoma. Blocking oxidative respiration and mTORC1 signaling abrogates the high rates of tumor cell proliferation and tumor growth specifically in SETD2-deficient tumors. Our data nominate SETD2 deficiency as a functional marker of sensitivity to clinically actionable therapeutics targeting oxidative respiration and mTORC1 signaling.

[1] Department of Cancer Biology, Perelman School of Medicine, University of Pennsylvania, Philadelphia, PA, USA. [2] Cell and Molecular Biology Graduate Program, Perelman School of Medicine, University of Pennsylvania, Philadelphia, PA, USA. [3] Center for Mitochondrial and Epigenomic Medicine, The Children's Hospital of Philadelphia, Philadelphia, PA, USA. [4] Abramson Family Cancer Research Institute, University of Pennsylvania, Philadelphia, PA, USA. [5] Present address: Dana-Farber Cancer Institute, Boston, MA, USA. [6] These authors contributed equally: David M. Walter, Amy C. Gladstein. ✉email: dfeldser@upenn.edu

nactivation of *SETD2* (SET domain containing 2, histone lysine methyltransferase) is a prevalent feature of many cancer types including ~7% of lung adenocarcinomas[1–3]. SETD2 has the unique catalytic activity for histone H3 lysine 36 trimethylation (H3K36me3) which it deposits along gene bodies during transcriptional elongation[4]. However, additional non-histone substrates have been identified which link SETD2 action with diverse roles in chromosome segregation (α-tubulin), interferon signaling (STAT1), or regulation of other chromatin modifying enzymes (EZH2)[5–7]. Loss of H3K36me3 due to SETD2 inactivation is implicated in the impairment of DNA repair, accurate splice site usage, transcriptional control, and DNA and RNA methylation[8–12]. These numerous characterized functions obscure how SETD2 mediates lung tumor suppression and whether inactivation of SETD2 presents therapeutic vulnerabilities that can be targeted to limit cancer growth.

To study *Setd2* inactivation in vivo, we previously used the well-characterized, Cre-inducible *Kras^LSL-G12D/+* (K) and *Kras^LSL-G12D/+; p53^flox/flox* (KP) mouse models of lung adenocarcinoma. Combined with a lentiviral vector that expresses Cre recombinase, along with the essential CRISPR components, Cas9 and an sgRNA to *Setd2*, we and others have shown that inactivation of *Setd2* in *Kras^G12D*-driven lung adenocarcinoma promotes cellular proliferation very early after tumor initiation; an effect that is widespread amongst all developing tumors[13,14]. Moreover, *Setd2* quantitatively ranks at the apex of all major tumor suppressors for its ability to suppress KRAS-driven cell proliferation[13,15]. However, unlike other tumor suppressors such as *Trp53* or *Rb1*, whose inactivation promotes cell state changes that drive malignant progression, loss of differentiation, and metastasis, inactivation of *Setd2* seems only to fuel cellular proliferation in these models[14,16,17]. As such, we sought to interrogate the consequences of *Setd2* inactivation in KRAS-driven lung adenocarcinoma in order to better understand how SETD2 deficiency drives early and widespread tumor growth and identify intrinsic therapeutic vulnerabilities that may exist.

## Results

### SETD2 deficiency promotes OXPHOS and protein synthesis gene expression programs

To gain mechanistic insights into how SETD2 constrains cancer growth, we performed a comprehensive analysis of RNA-sequencing data across multiple human cancer types expressing low or high levels of *SETD2*[2]. Using Gene Set Enrichment Analysis (GSEA), we identified an enrichment of ribosomal- and mitochondrial-associated gene sets that negatively correlated with *SETD2* expression. This association was strongly apparent in multiple tumor types that have a high frequency of *SETD2* mutations, in addition to lung adenocarcinoma (Fig. 1A and Supplementary Fig. 1A–D)[18]. The negative correlation of ribosomal- and mitochondrial-associated gene sets was unique to SETD2 expression, as these genes sets were not associated with expression levels of other major tumor suppressors in lung adenocarcinoma such as *TP53* or *RB1*, which instead correlated with DNA replication-associated gene sets (Supplementary Fig. 1E–G). These results suggest that ribosomal- and mitochondrial-biosynthetic pathways are activated in SETD2-deficient tumors and may be responsible for driving tumor cell proliferation. To extend this analysis, we analyzed gene expression data from *Kras^LSL-G12D/+* (K) tumors initiated with a non-targeting control lentiviral CRISPR vector (K-Ctrl) or an sgRNA targeting *Setd2* (K-Setd2^KO), as well as more advanced tumors isolated from *Kras^LSL-G12D/+; p53^flox/flox; YFP^flox/flox* (KPY-Ctrl or KPY-Setd2^KO) mice. In each case (K and KPY), control tumors and *Setd2^KO* tumors were assessed via histological analysis to select stage matched specimens for comparison. K-Setd2^KO tumors, which

were low grade adenomas, and KPY-Setd2^KO tumors, which were higher grade adenocarcinomas, had vastly different gene expression profiles than their K-Ctrl and KPY-Ctrl counterparts and a strong enrichment of ribosomal- and mitochondrial-associated gene sets. (Fig. 1B, C, Supplementary Fig. 1H, I). Collectively these results suggest that ribosomal- and mitochondrial-biosynthetic pathways are activated in SETD2-deficient tumors and may be responsible for driving increased tumor cell proliferation.

### SETD2-deficient tumors have altered mitochondrial morphology and function

The major driver of enrichment for mitochondrial-associated gene sets in SETD2-deficient tumors was the significant up-regulation of genes encoding mitochondrial electron transport chain (ETC) proteins in complexes I, III, IV and V (Fig. 1D). To determine whether physical changes occur in the mitochondria when *Setd2* is inactivated, we imaged KP-Ctrl and KP-Setd2^KO tumors by transmission electron microscopy. Although the overall frequency of mitochondria observed per field of view was similar between genotypes (Supplementary Fig. 2A, B), the mitochondria in KP-Setd2^KO tumors were significantly different than those found in KP-Ctrl tumors by several parameters. Grossly, the mitochondria of KP-Setd2^KO tumors were significantly smaller and had a more electron dense matrix than controls (Fig. 2A–C Supplementary Fig. 2A). Morphologically, KP-Setd2^KO mitochondria also had significantly more cristae overall, and on average each cristae was significantly more swollen (Fig. 2A, D, E)[18,19]. While some of these morphological changes can be associated with mitochondrial dysfunction and cell death, the higher electron density in the mitochondria matrix and a greater overall number of cristae in the mitochondria of KP-Setd2^KO tumors may suggest greater oxidative function[20].

To investigate mitochondrial properties in neoplastic cells directly ex vivo, we crossed a *Rosa26^LSL-YFP* Cre-reporter allele into the KP model, generating KPY mice (Supplementary Fig. 2C)[21]. Consistent with the EM data, KPY-Setd2^KO tumor cells had decreased staining for MitoTracker Deep Red indicative of decreased mitochondrial mass (Fig. 2F). Though KPY-Setd2^KO tumor cells did not differ from KPY-Ctrl tumor cells with respect to mitochondrial superoxide production, the mitochondrial oxidative potential and the total ETC activity was significantly higher in KPY-Setd2^KO tumor cells (Fig. 2G, Supplementary Fig. 2D, E).

To determine if the observed mitochondrial changes in SETD2-deficient tumors result in increased mitochondrial function and consequential ATP production, we generated cell lines expressing two distinct shRNAs targeting *Setd2* in the H2009 human lung adenocarcinoma cell line that harbors oncogenic *KRAS* and *p53* mutations[22]. As expected, *SETD2* shRNA-expressing cell lines had decreased SETD2 and H3K36me3 compared to control cell lines, confirming a knockdown of SETD2 (Supplementary Fig. 3A–D). To profile mitochondrial function we performed a cell mitochondrial stress test assay (Seahorse XF) on these cell lines. In agreement with our in vivo data, SETD2-deficient human cell lines displayed increased oxygen consumption rates and ATP production (Fig. 2H–K). These data demonstrate that SETD2-deficiency promotes increased mitochondrial metabolism and are consistent with the gene expression programs that are enriched in human and mouse tumors with low SETD2 expression.

### mTORC1 signaling and protein synthesis are heightened in SETD2-deficient tumors

An additional feature of SETD2-deficient tumors was the significant enrichment of genes associated with protein synthesis and ribosome biogenesis (Fig. 1B, C,

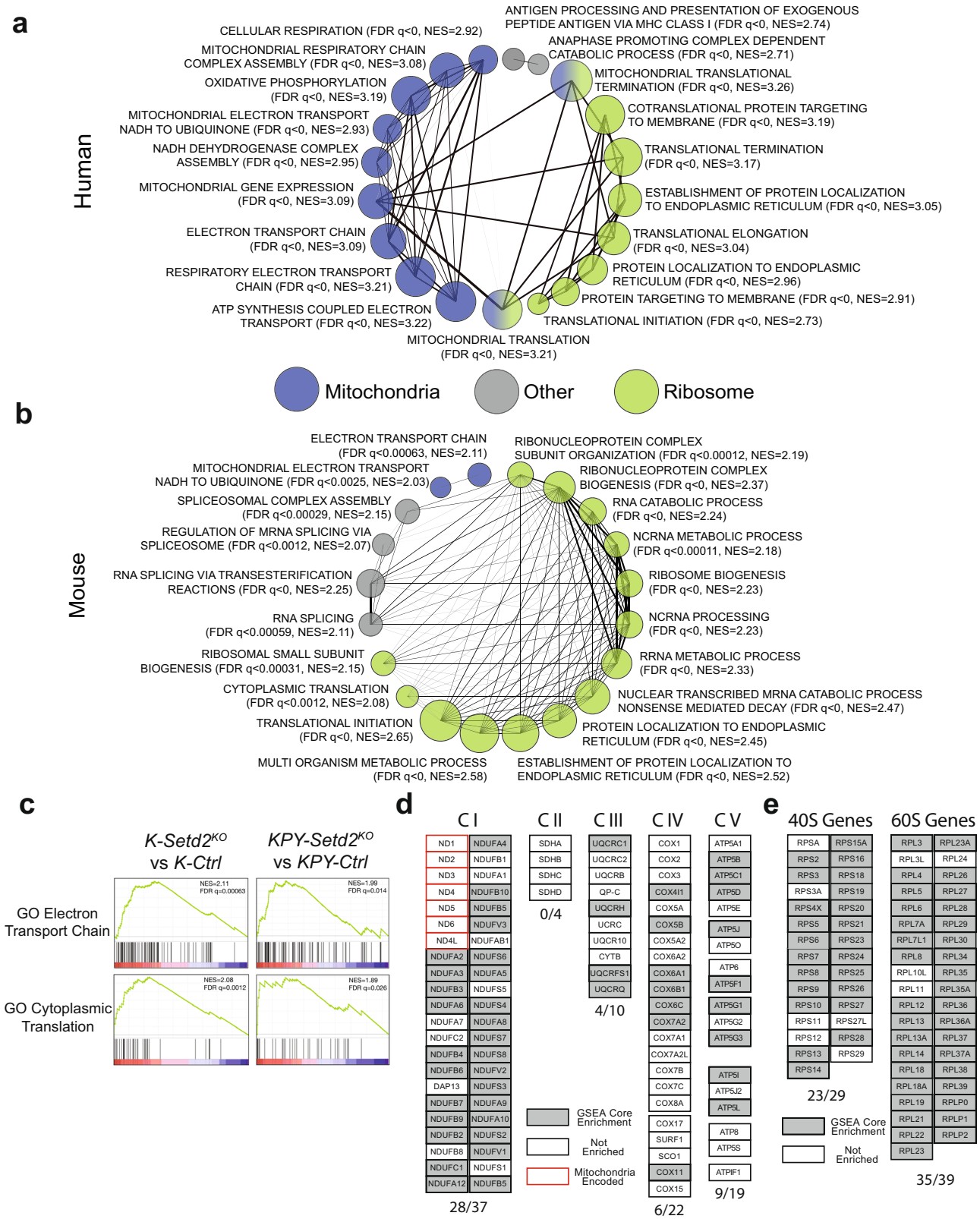

E)[23]. To assess the impact of SETD2 deficiency on protein synthesis, we pulse-labeled *K-Ctrl* and *K-Setd2*[KO] mice with the tRNA mimetic O-propargyl-puromycin (OP-Puro). OP-Puro is incorporated into actively translated proteins to allow for quantification of protein synthesis by flow cytometry or fluorescent microscopy and offers a method to quantify the rate of protein synthesis that is compatible with small sample sizes[24–26].

Combined staining for H3K36me3 and OP-Puro incorporation demonstrated a marked enhancement of overall protein synthesis in *K-Setd2*[KO] tumors (Fig. 3A, B). The incorporation of the *Rosa26*[LSL-YFP] allele in our *KPY* model afforded an orthogonal approach to study differences in protein synthesis rates by measuring the brightness of cells expressing the YFP reporter protein. While *KPY-Setd2*[KO] tumor cells were only slightly larger (3.8%

**Fig. 1 SETD2 deficiency promotes OXPHOS and protein synthesis gene expression programs in human and murine lung adenocarcinomas. a** Gene set enrichment analysis (GSEA) network plot of the 20 most enriched gene sets negatively-correlated with SETD2 expression in human lung adenocarcinomas. The size of each node corresponds to the normalized enrichment score (NES) and the width of the connecting lines indicates the number of overlapping genes between gene sets. Gene sets are categorized as mitochondrial, ribosomal or other according to the genes represented. **b** GSEA network plot of the top 20 enriched gene sets in *KPY-Setd2^KO* lung adenocarcinomas. The size of each node corresponds to the normalized enrichment score (NES) and the width of the connecting lines indicates the number of overlapping genes between gene sets. Gene sets are categorized as mitochondrial, ribosomal or other according to the genes represented. **c** Representative mitochondrial (GO Electron Transport Chain) and ribosomal (GO Cytoplasmic Translation) GO biological process ontology gene sets that are enriched in *K-Setd2^KO* (left) and *KPY-Setd2^KO* (right) tumors. **d** Depiction of mitochondrial electron transport chain genes that show enrichment in the molecular signatures database (mSigDB) Mootha VOXPHOS gene set in *K-Setd2^KO* tumors[53]. Genes are grouped according to the relevant ETC complex. Genes that are part of the core enrichment of the gene set are marked in gray, genes that are not enriched are marked in white, and mitochondrial encoded genes are outlined in red. **e** Depiction of ribosomal 40 S and 60 S genes that show enrichment in the mSigDB GO Ribosome Biogenesis gene set in *K-Setd2^KO* tumors. Genes that are enriched in this gene set are marked in gray while genes that are not enriched are marked in white.

higher FSC-A) and mRNA expression from the *Rosa26^LSL-YFP* allele was similar to *KPY-Ctrl* tumor cells, *KPY-Setd2^KO* tumor cells had an 18.5% increase in mean YFP fluorescence. This indicates a positive effect on YFP protein synthesis without affecting YFP mRNA production. (Supplementary Fig. 4A, B). These data suggest that SETD2 normally constrains the rate of protein synthesis in addition to the degree of OXPHOS. These functions, which would be expected to limit cellular proliferation, are consistent with the proliferation-driving effects of *Setd2* inactivation in KRAS-driven lung cancer.

Tightly linked with both protein synthesis and OXPHOS is the master nutrient sensing complex mTORC1[27–29]. To determine whether SETD2 deficiency is associated with increased mTORC1 activity, we first evaluated human lung adenocarcinomas using reverse phase protein array (RPPA) data from the Cancer Proteome Atlas[30,31]. An established marker of mTORC1 activity is the sequential phosphorylation of the translational repressor 4E-BP1, first at Thr37/Thr46 to prime subsequent phosphorylation at Thr70 and Thr65[32,33]. SETD2-deficient human tumors had significantly increased phosphorylation of 4E-BP1 at Thr70, while total 4E-BP1 levels were unchanged (Supplementary Fig. 4C). Additionally, there was a significant negative correlation between *SETD2* mRNA expression and phosphorylated 4E-BP1(T70) (Supplementary Fig. 4D). By profiling human lung adenocarcinoma genomic data we also found that tumors with low SETD2 expression were significantly enriched for an mTORC1 signaling-related gene set (Supplementary Fig. 4E)[2]. Consistent with these analyses of human datasets, we identified significantly increased mTORC1-dependent 4E-BP1(T37/46) phosphorylation in *K-Setd2^KO* tumors compared to *K-Ctrl* tumors (Fig. 3C, D)[34]. Further, SETD2-deficient tumors had significantly higher levels of mTORC1 localized at the lysosome where it is known to actively signal, further indicating that SETD2 deficiency promotes mTORC1 activity (Fig. 3E, F).

**Therapeutic targeting of OXPHOS and mTORC1 counteracts SETD2-deficient tumor growth.** Our discovery that mTORC1 signaling, protein synthesis, and mitochondrial OXPHOS are increased in SETD2-deficient tumors suggested that these processes may drive cell proliferation and thus offer therapeutic susceptibilities for *SETD2*-mutant cancers. To assess this possibility, we treated mice bearing established *K-Ctrl* and *K-Setd2^KO* tumors daily for 4 weeks with either the mTORC1 inhibitor rapamycin, the mitochondrial complex I inhibitor IACS-10759, or the anti-diabetic biguanide phenformin which inhibits both complexes (Fig. 4A)[35–37]. All three therapeutics had little effect on *K-Ctrl* tumors. However, each treatment significantly suppressed the increased tumor growth of *K-Setd2^KO* tumors (Fig. 4B, C). SETD2-deficient tumors were particularly sensitive to rapamycin treatment, which had a

similar suppressive effect on tumor growth as phenformin which inhibits both mitochondrial complex I and mTORC1 signaling (Fig. 4B, C). Inhibition of mTORC1 and mitochondrial complex I activity resulted in significantly reduced cell proliferation, marked by decreased phospho-H3 presence, demonstrating that the proliferative impact of SETD2 inactivation is driven, at least in part, through these pathways (Fig. 4D, Supplementary Fig. 5A). Further, inhibition of mTORC1 and mitochondrial complex I, alone or in combination, did not result in significant levels of cell death indicating that cell death is not the cause of the decreased tumor growth observed (Supplementary Fig. 5B). Phenformin is a highly potent inhibitor of both mitochondrial complex I and mTORC1. This potency has led to significant toxicity and its clinical replacement for the treatment of type II diabetes with the related biguanide metformin, which is used widely and is epidemiologically associated with suppressing cancer incidence[38–40]. Therefore, we treated *K-Ctrl and K-Setd2^KO* mice bearing established tumors with metformin for an extended period of 12 weeks, mimicking long-term metformin treatment. While metformin treatment had no impact on *K-Ctrl* tumor growth, metformin-treated *K-Setd2^KO* tumors were significantly smaller and less proliferative than vehicle treated *K-Setd2^KO* tumors (Fig. 4E–G).

**Discussion**

Here, we identify a conserved enhancement of gene expression programs and functional markers associated with oxidative metabolism and protein synthesis in SETD2-deficient cancers. The tumor suppressive function of the chromatin modifier SETD2, at least in the context of KRAS-driven lung adenocarcinoma, is therefore to limit pro-proliferative metabolic pathways by restricting oxidative metabolism and protein synthesis. As such, our data bolster the expanding realization that the metabolic state of a cell is intimately linked to the repertoire of post translational modifications on histones[41–43].

Our work uncovers a role for SETD2 in constraining mitochondrial OXPHOS and mTORC1 signaling to limit cellular proliferation in the context of KRAS-driven lung adenocarcinoma. It is notable that these roles are not entirely dissimilar to those regulated by the LKB1 tumor suppressor, a well-known regulator of nutrient-sensing mechanisms that impinge on mTORC1 signaling[44]. Further, silencing or inactivation of *Lkb1* also sensitizes lymphoma and lung adenocarcinoma to phenformin[35,45]. The parallels between the consequences of SETD2 and LKB1 inactivation are all the more provocative given an apparent functional-genetic epistatic relationship in KRAS-driven mouse models and their mutually exclusive pattern of mutation in human of lung adenocarcinoma patients[2,15]. Determining whether and how LKB1 and SETD2 fit into a singular

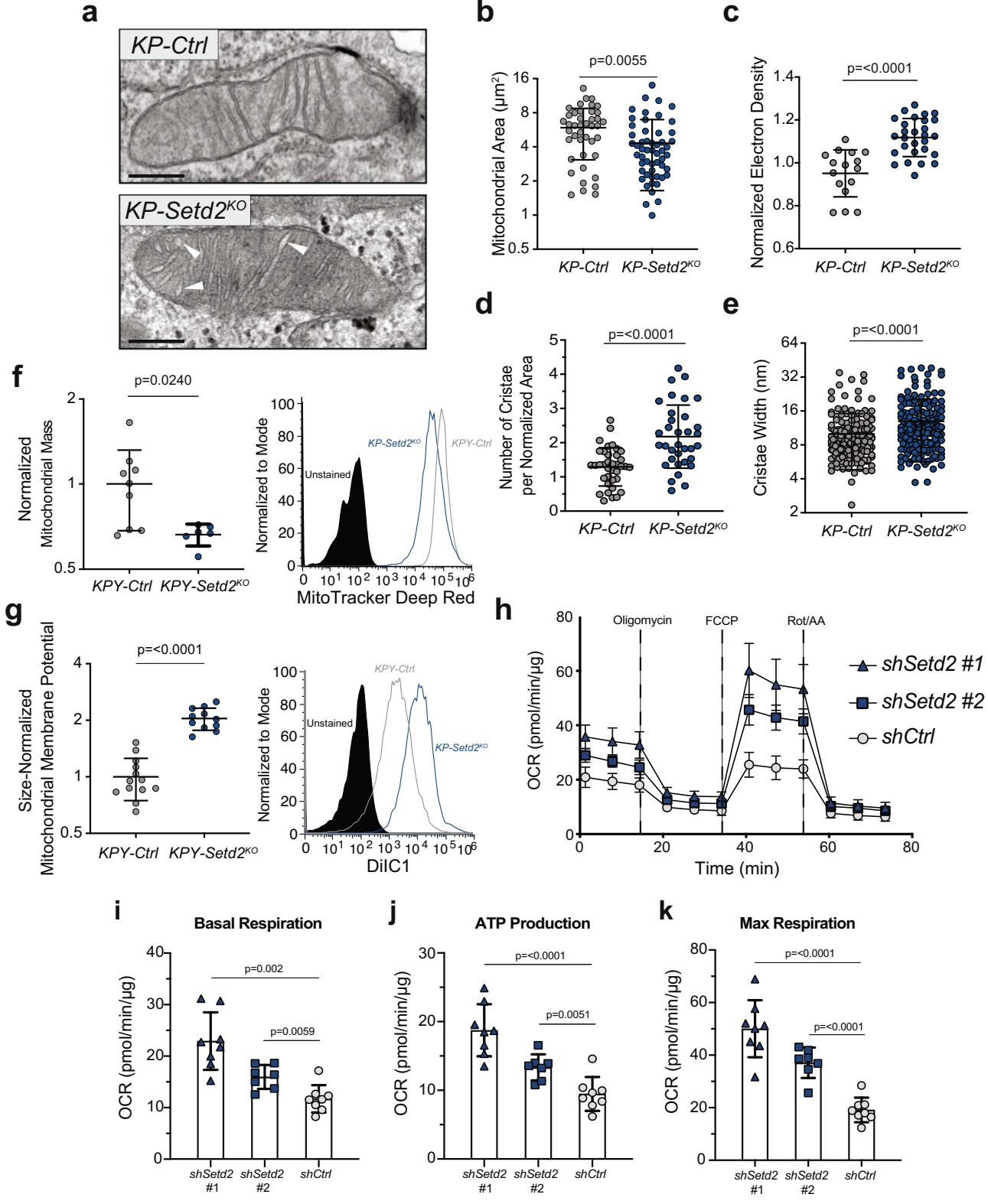

pathway predicted by these overlapping phenotypes and their genetic epistasis is an intriguing possibility that has yet to be determined.

Finally, we demonstrate not only that signaling through OXPHOS and mTORC1 is required for the proliferative benefit bestowed upon tumor cells following SETD2 inactivation, but that they also represent readily actionable therapeutic vulnerabilities for patients with SETD2-deficient tumors. Therefore, our

study nominates OXPHOS and mTORC1 inhibition as a targeted therapy for SETD2-deficient lung adenocarcinoma.

## Methods

**Animal studies and treatment**. Animal studies were performed under strict compliance with Institutional Animal Care and Use Committee at University of Pennsylvania (804774). $Kras^{LSL-G12D}$ mice (Jax stock number 008179), $Trp53^{flox/flox}$ mice (Jax stock number 008462), and $Rosa26^{LSL-YFP/ LSL-YFP}$ mice have

**Fig. 2 SETD2-deficient tumors contain mitochondria with distinct morphological features and increased electron transport chain activity.**
**a** Transmission electron micrograph of mitochondria from a *KP-Ctrl* tumor (left) or *KP-Setd2^KO* tumor (right). White arrows denote swollen mitochondrial cristae. Scale bars = 250 nm. **b** Quantification of the area of individual mitochondria measured from electron micrographs. Data indicate the mean ± standard deviation. Data points represent individual mitochondria (*KP-Ctrl*: n = 42 mitochondria, n = 3 mice, *KP-Setd2^KO*: n = 54 mitochondria, n = 3 mice). Significance determined by unpaired Student's *t*-test. **c** Quantification of the matrix electron density of individual mitochondria measured from electron micrographs. Data indicate the mean ± standard deviation. Data points represent individual mitochondria (*KP-Ctrl*: n = 16 mitochondria, n = 2 mice, *KP-Setd2^KO*: n = 28 mitochondria, n = 1 mice). Significance determined by unpaired Student's *t*-test. **d** Quantification of the number of cristae per mitochondria normalized to mitochondrial area. Data indicate the mean ± standard deviation. Data points represent individual mitochondria (*KP-Ctrl*: n = 37 mitochondria, n = 3 mice, *KP-Setd2^KO*: n = 34 mitochondria, n = 3 mice). Significance determined by unpaired Student's *t*-test. **e** Quantification of the width of individual mitochondrial cristae measured from electron micrographs. Data indicate the mean ± standard deviation. Data points represent individual cristae (*KP-Ctrl*: n = 179 cristae, n = 42 mitochondria, n = 3 mice, *KP-Setd2^KO*: n = 177 cristae, n = 50 mitochondria, n = 3 mice). Significance determined by unpaired Student's *t*-test. **f** Quantification of the total mitochondrial mass by median fluorescence intensity of MitoTracker Deep Red FM within tumor cells isolated from *KPY* mice. Data represent the mean ± standard deviation. Data points represent individual tumors (*KPY-Ctrl*: n = 9 tumors, n = 2 mice, *KPY-Setd2^KO*: n = 6 tumors, n = 2 mice). Significance determined by unpaired Student's *t*-test. Histogram shows representative flow data from *KPY*-Ctrl and *KPY-Setd2^KO* tumors with unstained control. **g** Quantification of the mitochondrial membrane potential by median fluorescence intensity of MitoProbe DiIC1(5) within tumor cells isolated from *KPY* mice. Data is normalized to the mitochondrial mass of each sample. Data indicate the mean ± standard deviation. Data points represent individual tumors (*KPY-Ctrl*: n = 13 tumors, n = 2 mice, *KPY-Setd2^KO*: n = 11 tumors, n = 2 mice). Significance determined by unpaired Student's *t*-test. Histogram shows representative flow data from *KPY*-Ctrl and *KPY-Setd2^KO* tumors with unstained control. **h** Seahorse XF cell mitochondrial stress test assay performed in H2009 sh*Setd2* and sh*Ctrl* cells. Relative oxygen consumption rate was normalized to total protein abundance. Each symbol in OCR profile plots represents the mean of at least n = 6 technical replicates of three reading cycles. OCR profile plots for (**i**) basal respiration (**j**) ATP production, and (**k**) maximal respiration. Each symbol represents one technical replicate per cell line. Data indicate the mean ± standard deviation. Significance determined by unpaired Student's *t*-test.

previously been described[21,46,47]. Mice are mixed B6J/129S4vJae. Mice were transduced with 6×10⁴ plaque forming units (PFUs) per mouse of Lenti-CRISPRv2Cre by endotracheal intubation[48]. LentiCRISPRv2Cre expressing sgRNAs targeting *GFP* or *β-Galactosidase* (BGal) as controls, while an sgRNA targeting *Setd2* was used for knockouts. For control sgRNAs, sgGFP was used to induce *K* tumors for RNA-sequencing, while sgBGal was used for all other experiments. The sgRNA sequences are: sgGFP-GGGCGAGGAGCTGTTCACCG, sgBGal – CACGTAGATACGTCTGCATC, and sgSetd2-AATGGGCTGAGGTAC GCCGT[14,49]. Lentivirus production and titration was performed as described previously[14].

For drug experiments, OpenStandard Diet was formulated with Rapamycin (MedChem Express) at 15 mg kg⁻¹ for a dose of 2 mg/kg/day or IACS-10759 (MedChem Express) at 37.5 mg kg⁻¹ for a dose of 5 mg/kg/day by Research Diets. Mice were placed on treatment diet 4 weeks prior to analysis. For phenformin treatment, mice were given the drug by oral gavage daily on a 5 days on/2 days off schedule for 4 weeks. Mice were given 200 mg/ kg/day phenformin dissolved in water (Cayman Chemicals). For metformin treatment, metformin was placed in the drinking water of mice at 1.25 mg/ml for 12 weeks prior to sacrifice (Sigma-Aldrich). No statistical methods were used to predetermine sample sizes. The size of each animal cohort was determined by estimating biologically relevant effect sizes between control and treated groups and then using the minimum number of animals that could reveal statistical significance using the indicated tests of significance. All animal studies were randomized in 'control' or 'treated' groups, and roughly equal proportions of male and female animals were used. However, all animals housed within the same cage were generally placed within the same treatment group. For histopathological assessments of tumor size, researchers were blinded to sample identity and group. The animal protocol was approved by the University Laboratory Animal Resources (ULAR) at the University of Pennsylvania and the IACUC.

**Immunohistochemistry and immunofluorescence.** Lung and tumor tissues were dissected into 10% neutral-buffered formalin overnight at room temperature before dehydration in a graded alcohol series. Paraffin-embedded and H&E-stained histological sections were produced by the Penn Molecular Pathology and Imaging Core. Immunostaining for H3K36me3 (Abcam, ab9050, 1:1000), p-4E-BP1(T37/46) (Cell Signaling Technology, cs2855, 1:100), mTOR (Cell Signaling Technology, cs2983, 1:100), LAMP2 (Abcam, ab13524, 1:100) and p-H3 (Cell Signaling Technology, cs9701, 1:500) were performed after citrate-based antigen retrieval. H3K36me3 alone was assessed by immunohistochemistry using ABC reagent (Vector Laboratories, PK-4001) and ImmPACT DAB (Vector Laboratories, SK-4105) according to the product instructions. P-4E-BP1(T37/46) was assessed by immunofluorescence using a biotinylated secondary antibody (Vector Laboratories, PK-4001) according to product instructions, and Streptavidin-conjugated Alexa594 (Thermo Fisher S11227, 1:200). Colocalization of mTOR and LAMP2 was determined using an anti-Rat Alexa647 antibody (Thermo Fisher A21247, 1:200) to detect LAMP2, and a biotinylated anti-Rabbit secondary antibody (Vector Laboratories, PK-4001) followed by Streptavidin-conjugated Alexa488 (Thermo Fisher S32354, 1:200) to detect mTOR.

Immunohistochemistry and immunofluorescence were both performed on paraffin-embedded sections following the same antigen-retrieval protocol. Sections were incubated in primary antibody overnight at 4 °C, secondary antibody for 1 hour at room temperature, and for immunofluorescence Streptavidin-conjugated fluorophore for 1 hour at room temperature in the dark.

For TUNEL staining, tissues were deparaffinized and then permeabilized with 0.1% sodium citrate and 0.1% Triton-X in PBS for 8 minutes. FITC-conjugated TUNEL labeling mix (Millipore Sigma, 11684795910) was added to permeabilized tissue sections and incubated for 1 hour at 37 °C in the dark. For all immunofluorescence staining, nuclei were stained using 5 mg/ml DAPI at a 1:1000 dilution for 10 minutes, and then slides were mounted with Fluoro-Gel (EMS, 17985-50).

**O-Propargyl-Puromycin Analysis.** For the quantification of protein translation, mice were injected intraperitoneally (IP) with 200 μl of a 10 mM solution of OP-Puro dissolved in PBS as previously described[24]. 1 hour after OP-Puro IP injection, mice were sacrificed and lungs were formalin fixed and paraffin embedded as described above. Co-immunofluorescence for H3K36me3 and OP-Puro was performed to quantify translation rates in tumors lacking SETD2 activity. Antigen retrieval was performed using a solution of 20 μg/ml proteinase K in TE Buffer (pH 8) at 37 °C for 10 minutes. A click chemistry reaction was then performed for 30 minutes at room temperature in the dark to conjugate Alexa594 to incorporated OP-Puro according to product instructions (Thermo Fisher, C10429, beginning at Step 5.1). Samples were kept in the dark for all further steps. Samples were treated with avidin and biotin blocking steps for 20 minutes each (Vector Laboratories, SP-2001), and a 30-minute protein block (Dako, X090930-2) before incubating with H3K36me3 primary antibody (Abcam, ab9050) at 1:300 overnight at 4 °C. H3K36me3 was then detected using a biotinylated secondary antibody for 1 hour (Vector Laboratories, PK-4001) followed by streptavidin-conjugated Alexa488 for 1 hour at a 1:200 dilution (Thermo Fisher, S-32354). Nuclei were stained using 5 mg/ml DAPI at a 1:1000 dilution for 10 minutes, and then slides were mounted with Fluoro-Gel (EMS, 17985-50).

**Histological quantification.** The analysis of mTOR/LAMP2 colocalization was performed using a Leica TCS SP5 II confocal microscope. Z-stack projections of confocal images taken of control and SETD2-deficient tumors were analyzed. For the quantification of PLA individual loci were counted from z-stack projections. The LAS X colocalization tool was used for the quantification of mTOR/LAMP2 colocalization, All other photomicrographs were captured on a Leica DMI6000B inverted light and fluorescence microscope, and ImageJ software was used for subsequent histological quantifications[50]. For the quantification of staining intensity for OP-Puro and p-4EBP1(T37/46), single fluorescent channel images were obtained and the mean fluorescence intensity of staining for each tumor was quantified in ImageJ. Great care was made to ensure that background signal from blood vessels, or empty spaces were excluded from the analysis. For the quantification of tumor sizes under varying conditions including drug treatments, a tile scan of each mouse lung was obtained using the Leica DMI600B microscope and tumor area was measured in ImageJ. Tumor areas were then normalized to the mean area of a sgCtrl, vehicle treatment tumor. For the quantification of p-H3 staining, total nuclei in each tumor were counted using the IHC Profiler plugin for ImageJ, and p-H3-expressing nuclei

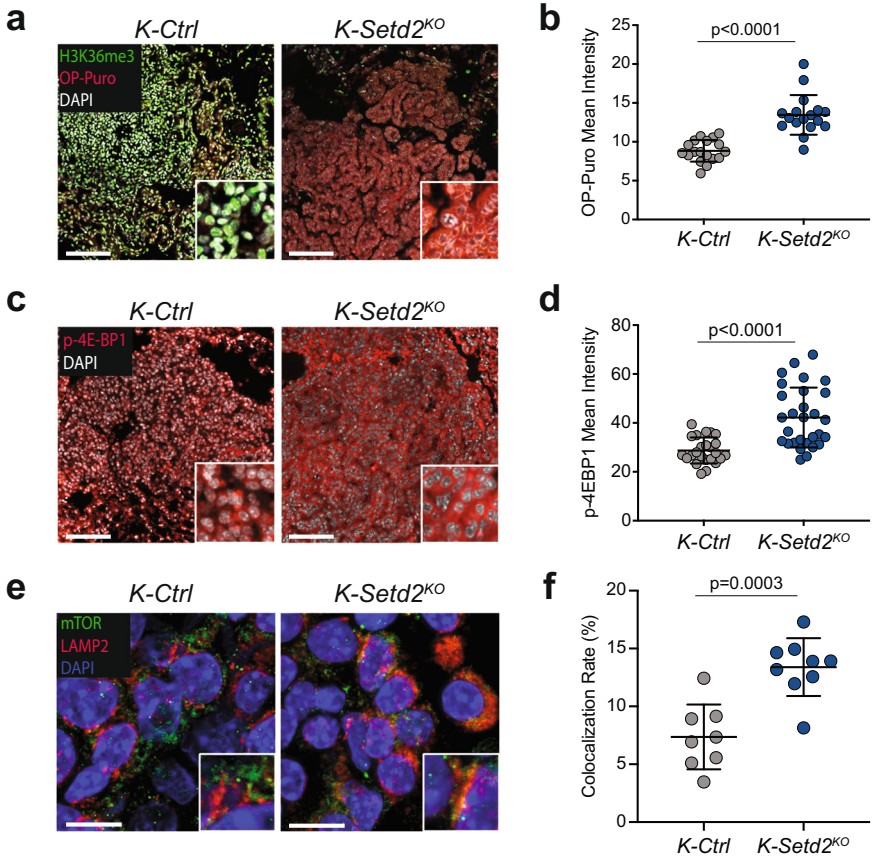

**Fig. 3 SETD2-deficient tumors have increased protein synthesis and high mTORC1 signaling. a** Representative images of o-propargyl-puromycin (OP-Puro) incorporation (red), H3K36me3 (green), and DAPI-stained nuclei (white) in *K-Ctrl* (left) and *K-Setd2$^{KO}$*(right) tumors. Scale bars = 100 μm, insets are magnified 5x. **b** Quantification of OP-Puro mean fluorescence intensity in *K-Ctrl* and *K-Setd2$^{KO}$* tumors. Data represent the mean ± standard deviation. Data points represent individual tumors (*K-Ctrl*: $n = 18$ tumors, $n = 3$ mice, *K-Setd2$^{KO}$*: $n = 17$ tumors, $n = 3$ mice). Significance determined by unpaired Student's $t$-test. **c** Representative images of phosphorylated 4E-BP1(T37/46) staining (red) in *K-Ctrl* (left) and *K-Setd2$^{KO}$*(right) tumors. Nuclei are counterstained with DAPI (white). Scale bars = 100 μm, insets are magnified 5x. **d** Quantification of p-4E-BP1(T37/46) mean fluorescence intensity in *K-Ctrl* and *K-Setd2$^{KO}$* tumors. Data represent the mean ± standard deviation. Data points represent individual tumors (*K-Ctrl*: $n = 24$ tumors, $n = 3$ mice, *K-Setd2$^{KO}$*: $n = 29$ tumors, $n = 3$ mice). Significance determined by unpaired Student's $t$-test. **e** Representative images of co-immunofluorescence of mTOR (green) and LAMP2 (red) indicating localization of mTOR at the lysosome (yellow) in *K-Ctrl* (left) and *K-Setd2$^{KO}$*(right) tumors. Nuclei are counterstained with DAPI (blue). Scale bars = 10 μm, inset is magnified 2X. **f** Quantification of the percentage of colocalization between mTOR and LAMP2 in *K- Ctrl* and *K-Setd2$^{KO}$* tumors. Data points represent individual tumors (*K-Ctrl*: $n = 8$ tumors, $n = 3$ mice, *K-Setd2$^{KO}$*: $n = 9$ tumors, $n = 3$ mice). Significance determined by unpaired Student's $t$-test.

were counted in ImageJ using the Cell Counter plugin[51]. For all histological analyses each data point represents an individual tumor.

**Flow Cytometry**. Tumors were microdissected directly from the lungs of *Kras$^{LSL-G12D/+}$;Trp53$^{flox/flox}$; Rosa26$^{LSL-YFP/LSL-YFP}$* (*KPY*) mice and individually placed in 500 μl of tumor digestion buffer consisting of PBS containing 10 mM HEPES pH 7.4, 150 mM NaCl, 5 mM KCl, 1 mM MgCl₂, and 1.8 mM CaCl₂, along with freshly added Collagenase 4 (Worthington 100 mg/ml solution, 20 μl per ml of digestion buffer) and DNase I (Roche 10 mg/ml solution, 4 μl per ml of digestion buffer). Tumors were manually disassociated using scissors, and then placed in a 4 °C shaker for 1 hour at 250 rpm. Digested tumors were then filtered into strainer-cap flow tubes (Corning, 352235) containing 1 ml of horse serum (Thermo Fisher, 16050122) to quench the digestion reaction. Cells were spun down at 200 g for 5 minutes with the cap in place to obtain all cells. The supernatant was aspirated, cells were washed once with PBS and then resuspended with a given mitochondrial dye to stain for 30 minutes at 37 °C. To quantify mitochondrial volume, cells were incubated with 50 nM MitoTracker Deep Red FM (Thermo Fisher, M22426), combined with 50 μM of CCCP (Thermo Fisher, M34151) to eliminate confounding effects of mitochondrial membrane potential differences. To quantify mitochondrial membrane potential cells were incubated with 20 nM MitoProbe DiIC1(5) (Thermo Fisher, M34151). To quantify mitochondrial oxidative potential cells were incubated with 100 nM MitoTracker Red CM-H₂XRos (Thermo Fisher, M7513). To quantify mitochondrial ROS cells were incubated with 5 μM MitoSOX Red (Thermo Fisher, M36008). After 30 minutes of staining, cells were washed twice with PBS and then resuspended in 100 μl of staining solution of FACS buffer

containing biotinylated antibodies against CD31 (BD Biosciences, 558737, 1:100), CD45 (BD Biosciences, 553078, 1:200), and Ter-119 (BD Biosciences, 553672, 1:100), for 25 minutes at a 4 °C. Cells were washed twice and resuspended in 100 μl of streptavidin-conjugated APC- eFluor 780 (Thermo Fisher, 47-4317-82) for 20 minutes. Finally, cells were washed twice and resuspended in FACS buffer containing DAPI at a 1:1000 dilution. Flow cytometry was performed using an Attune NxT flow cytometer (Thermo Fisher), and gating was performed to exclude doublets, dead cells, YFP- cells and non-epithelial contaminating cell types (see Supplementary Fig. 2c). Mitochondrial properties were then quantified by measuring the median fluorescence intensity of a given dye in live, YFP + tumor cells. For the quantification of YFP protein expression the mean fluorescence intensity of YFP was quantified for each sample by flow cytometry. For all flow cytometry analyses each data point represents an individual tumor.

**RNA sequencing and human dataset analysis**. For gene expression analysis in human tumors, RNA-sequencing data was obtained from the Cancer Genome Atlas lung adenocarcinoma dataset[2]. Due to the relative infrequency of *SETD2* mutations, a comparison of mutant and wildtype cases had insufficient statistical power to draw meaningful conclusions. Therefore, mRNA expression data was extracted by the Penn Institute for Biomedical Informatics, and a pearson correlation score was calculated comparing the expression of *SETD2* to all other genes. All genes were ranked in order according to the genes most negatively correlated with *SETD2* expression to most positively correlated, and this rank list was used to perform Gene Set Enrichment Analysis (GSEA) examining GO biological processes using the molecular signature database (MSigDB)[52,53]. For gene expression analysis

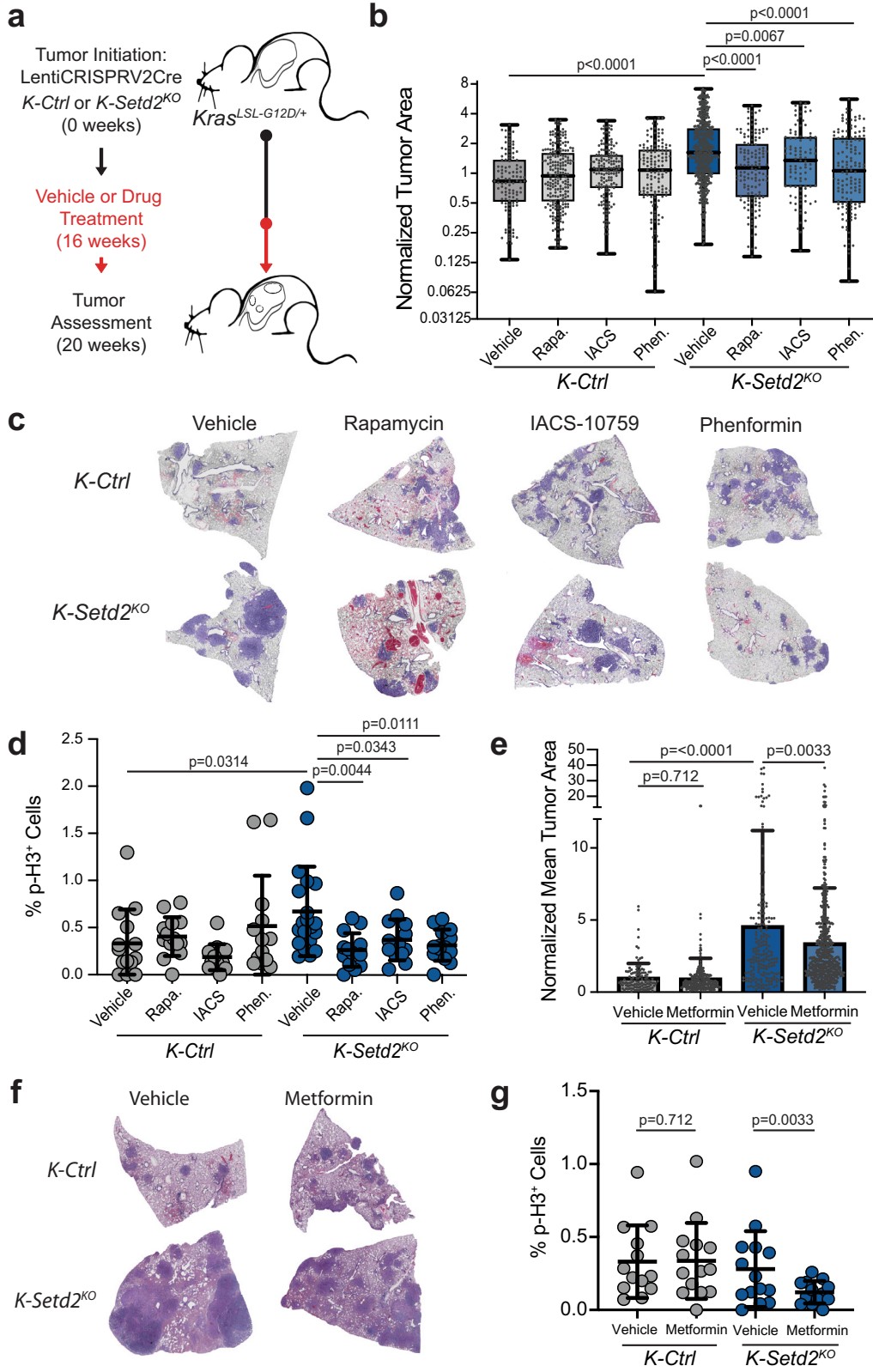

in *K-Ctrl* and *K-Setd2^KO* tumors, analysis was performed on previous sequencing results[14]. For gene expression analysis in *KPY-Ctrl* and *KPY-Setd2^KO* tumors, tumors were microdissected away from normal lung tissue, and digested into a single cell suspension as described above. Live, YFP + tumor cells were isolated by cell sorting, spun down and flash frozen in liquid nitrogen. RNA was extracted using the RNeasy Plus Micro kit (Qiagen, Catalog #74034) using 350 µl of RLT Plus and the QIAshredder columns as per manufacturer's instructions. Total RNA quantity was measured using the Qubit RNA HS assay kit (ThermoFisher, Catalog

#Q32852) and RNA quality was measured using a BioAnalyzer RNA 6000 Nano assay (Agilent, Catalog #5067-1511). Sequencing libraries were prepared on the Illumina NeoPrep and subjected to 75-bp single-end sequencing on the Illumina NextSeq 500 platform. Fastq files for each sample were aligned against the mouse genome, build GRCm38.p5, using Salmon (v0.8.2)[54].

Differentially expressed genes were identified with DESeq2 (v1.17.0) and ranked according to the Stat value which considers both the significance, fold-change and directionality of the gene expression change (Supplementary Data 1)[55]. This rank

**Fig. 4 mTORC1 signaling and mitochondrial OXPHOS are required for the growth- promoting effects caused by SETD2-deficiency. a** Schematic of experiment whereby *K-Ctrl* or *Setd2KO* tumors were initiated using LentiCRISPRv2Cre. 16 weeks after tumor initiation (black line) mice were randomly assigned to vehicle or drug treatment regimen for 4 weeks (red line), at which point mice were sacrificed for assessment. For metformin treatment, mice were given the drug for 12 weeks prior to sacrifice. **b** Quantification of mean tumor areas in vehicle-, rapamycin-, IACS-10759-, or phenformin-treated *K-Ctrl* or *K-Setd2KO* mice. Individual tumor sizes were normalized to the mean area of vehicle-treated control mice. Box and whisker plots indicate the median, lower quartile, upper quartile, maximum and minimum data points. Data points represent individual tumors (*K-Ctrl*: $n = 109$ vehicle, $n = 216$ rapamycin, $n = 155$ IACS-10759, $n = 144$ phenformin; *K-Setd2KO*: $n = 503$ vehicle, $n = 142$ rapamycin, $n = 99$ IACS-10759, $n = 138$ phenformin). Significance determined by unpaired Student's *t*-test. **c** Representative scans of tumor-bearing lobes from *K-Ctrl* or *K-Setd2KO* mice treated with vehicle, rapamycin, IACS-10759 or phenformin. **d** Quantification of cell proliferation by the percentage of p-H3 positive cells in *K-Ctrl* or *K-Setd2KO* tumors treated with vehicle, rapamycin, IACS-10759 or phenformin. Data indicate the mean ± standard deviation. Data points represent individual tumors (*K-Ctrl*: $n = 14$ vehicle, $n = 14$ rapamycin, $n = 13$ IACS-10759, $n = 13$ phenformin; *K-Setd2KO*: $n = 20$ vehicle, $n = 14$ rapamycin, $n = 14$ IACS-10759, $n = 14$ phenformin). Significance determined by unpaired Student's *t*-test. **e** Quantification of mean tumor areas in vehicle- and metformin-treated *K-Ctrl* or *K- Setd2KO* mice. Individual tumor sizes normalized to the mean area of vehicle-treated control mice. Data represent the mean ± standard deviation. Data points represent individual tumors (*K-Ctrl*: $n = 115$ vehicle, $n = 236$ metformin; *K-Setd2KO*: $n = 193$ vehicle, $n = 503$ metformin). Significance determined by unpaired Student's *t*-test. **f** Representative scans of tumor-bearing lobes from *K-Ctrl* or *K-Setd2KO* mice treated with vehicle or metformin. **g** Quantification of cell proliferation by the percentage of p-H3 positive cells in *K-Ctrl* and *K-Setd2KO* mice given vehicle treatment or metformin. Data indicate the mean ± standard deviation. Data points represent individual patient tumors (*K-Ctrl*: $n = 13$ Vehicle, $n = 14$ Metformin; *K-Setd2KO*: $n = 14$ Vehicle, $n = 13$ Metformin). Significance determined by unpaired Student's *t*-test. Note: No survival experiments were conducted and efficacy of drug treatments is solely based on the differences in tumor size shown here.

of genes most upregulated upon *Setd2* loss was then used to perform GSEA examining GO biological processes using the MSigDB. GSEA network plots of the top 20 pathways negatively correlated with *SETD2* expression in both human and mice were then generated as previously described[23]. A graphical depiction of the network plot was then generated using Gephi v.0.9.2, and gene sets were characterized according to their functions[56]. To quantify YFP RNA expression the total YFP read count was quantified from RNA- sequencing data of *KPY* tumors using Salmon (v0.8.2).

To analyze 4E-BP1 protein levels and phosphorylation in human tumors, level 4 RPPA data from human lung adenocarcinomas was extracted from the Cancer Proteome Atlas website[30,31]. RPPA z-scores were matched with gene expression and genetic information from each sample represented in the TCGA lung adenocarcinoma dataset[2]. The RPPA z-score for phosphorylated 4E-BP1(T70) was compared to *SETD2* mRNA expression z-scores for each sample. The levels of total 4E-BP1 and phosphorylated 4E-BP1(T70) were also compared between tumor samples with wildtype *SETD2*, and *SETD2* deficiency (defined as tumors containing either an inactivating mutation in *SETD2*, homozygous loss of the gene, or a loss of 1 copy of *SETD2* along with a mRNA z-score $< -0.5$).

**Electron Microscopy.** Tumors were microdissected directly from the lungs of mice and then the tissue was divided in half. One portion of each tumor was fixed for IHC to determine the H3K36me3-status of the given tumor as described above, while the other portion was fixed overnight in an osmium solution obtained from the Penn Electron Microscopy Resource Lab (EMRL), and then submitted to the EMRL for further tissue processing and staining with uranyl acetate and lead citrate. Transmission electron microscopy (TEM) was then performed using a JEOL JEM-1010 for both control and SETD2- deficient tumor samples. Images were taken at 60,000 to 150,000 X magnification, and mitochondrial properties were then quantified using ImageJ, normalizing to the magnification of the image. Mitochondrial size was in Image J, while mitochondrial number was quantified by counting the number of mitochondria per field of view across multiple 15,000 X magnification images. Mitochondrial cristae width was quantified in Image J by drawing a perpendicular line between the inner membranes of cristae, and then quantifying the resulting distance. Mitochondrial electron density was quantified by measuring the mean pixel darkness of the mitochondrial matrix in ImageJ and normalizing this to the mean pixel darkness of the surrounding cytoplasm. Mitochondrial cristae density was quantified by counting the number of cristae in an individual mitochondrion and then dividing by the mitochondrial area.

**Cell lines.** SETD2 shRNA sequences from Skutcha et al.[57] were cloned into the retroviral MLP shRNA expression vector. shRNA sequences are: shSETD2 #1: CAAGCAAAGAAGTATTCAGAA and shSETD2 #2: CAACCAA-CAGTCTGTCAGTGT. NCI-H2009 human lung adenocarcinoma cells (from NCI cell line repository) were infected with retrovirus harboring shSETD2 #1, shSETD2 #2, or MLP empty vector. Antibiotic selection was performed and efficiency of SETD2 knockdown was assessed via immunoblot analysis and RT-PCR.

**Immunoblot analysis.** For whole cell lysates, cells were lysed in RIPA buffer. Acid-extracted histones were prepared for histone methylation Western blot. Samples were resolved on NuPage 4–12% Bis-Tris protein gels (Thermo Fisher) and transferred to polyvinylidene fluoride (PVDF) membranes. Blocking, primary and secondary antibody incubations were performed in Tris-buffered saline (TBS) with 0.1% Tween-20. H3K36me3 (1:1000, Abcam, ab9050), SETD2 (1:1000, Cell

Signaling Technology, E4W8Q), H3 (1:10000, Abcam ab1791), β-actin (1:10000,Sigma Aldrich, A2066), were assessed by western blotting. β-actin and H3 were used as loading controls. Protein concentration was determined using a BCA protein assay kit (Thermo Fisher Scientific, 23225).

**Quantitative reverse transcription-PCR.** Total RNA was extracted from cells using Qiagen RNeasy Mini Kit (Qiagen, 74106). cDNA synthesis was performed using High Capacity cDNA Reverse Transcription Kit (Applied Biosystems, 4368814). RT-PCR was performed using SYBR Green I Nucleic Acid Gel Stain (Invitrogen, S7563) in triplicate, following manufacturer instructions, and evaluated on an Applied Biosystems ViiA 7 RT-PCR machine. Setd2 forward primer: CTTCTACCACGTATCAGCAACC, *Setd2* reverse primer: GTAATCACGTGTC CCACCATAC. *β-actin* forward primer: CCAACCGCGAGAAGATGA, *β-actin* reverse primer: CCAGAGGCGTACAGGGATAG.

**Seahorse XF Cell Mito Stress Analysis.** Oxidative respiration was measured using XF Cell Mito Stress Test Kit (Agilent Technologies, 103015-100). $1 \times 10^4$ cells per well were seeded on an XF96 Cell Culture Microplate. Microplate was incubated for 24 h at 37 C. Seahorse XF96 FluxPak sensor cartridge was hydrated with 200 μl of Seahorse Calibrant in a non-CO2 incubator at 37 C overnight. After 24 h, cells were incubated with base medium (Agilent Technologies, 102353-100) containing 2 mM L-glutamine, 1 mM sodium pyruvate, and 10 mM glucose in a non-$CO_2$ incubator at 37 C for 45 min prior to assay. Oxygen consumption rate (OCR) was measured by XFe96 extracellular flux analyzer with sequential injections of 1 μM oligomycin, 1 μM FCCP, and 0.5 μM rotenone/antimycin A. After the run, cells were lysed with 15 μl RIPA buffer and protein concentration was quantified using Pierce BCA Protein Assay Kit (Thermo Fisher Scientific, 23225). OCR measurements were normalized to the protein concentration in each well.

**Statistics and Reproducibility.** All analyses were performed using Graphpad Prism (v.8.1.1). For all analyses of mitochondrial properties, immunofluorescence, percentage of p-H3$^+$ cells, normalized tumor areas and RPPA analysis comparing *SETD2* wildtype and deficient tumors, unpaired Student's *t*-tests were performed. For the comparison of *SETD2* mRNA expression and p-4E-BP1(T70) levels by RPPA, a linear regression analysis was performed. Outliers were excluded from rapamycin, IACS-10759 and phenformin experiments using the ROUT method with a Q of 0.1%. Sample sizes for individual experiments are indicated in the figure legends. Reproducibility of the findings were confirmed by analyzing at least $n = 3$ technical and/or independent biological replicates as indicated in the figure legends. The findings of all the biological replicates were consistent.

**Reporting summary.** Further information on research design is available in the Nature Portfolio Reporting Summary linked to this article.

## Data availability
The raw RNA sequencing data associated with Fig. 1b–e and Supplementary Fig. 1h, i have been deposited publicly in the Gene Expression Omnibus (GEO) under accession number GSE224270. The list of differentially expressed genes associated with Fig. 1b–e and Supplementary Fig. 1h, i are available in Supplementary Data 1. Source data associated with Figs. 2b–l, 3b, d, f, 4b, d, e, g and Supplementary Fig. 2b, d, e, 3b, d, and

4a–d are available in Supplementary Data 2. Uncropped western blots are available in Supplementary Fig. 6.

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

## Acknowledgements

We would like to thank ULAR staff for animal husbandry, the Molecular Pathology and Imaging Core (MPIC) for histological analysis, B. Zuo for help with electron microscopy, A. Bedenbaugh for tissue processing, A. Durham and E. Radaelli for help with pathology, Bang-Jin Kim for assistance with confocal microscopy, N. Anderson for help with

mitochondrial stress test assay, and M. Winslow, K. Wellen, and S. Zhao for helpful discussions and critical reading of the manuscript. This work is supported by: NIH grants (R01-CA262619 and R01-CA222503 to D.M.F., 2-T32-CA-15299-15 to A.C.G.) and Department of Defense grant (LCD400095 to D.M.F.).

## Author contributions

D.M.W. and A.A.G. performed animal studies. D.M.W., R.N., and A.C.G. performed bioinformatics analyses with supervision from I.A.A. and D.M.F. A.C.G. performed cell culture studies. D.M.W., K.R.D., K.M.A., and J.O.A. performed histopathological analyses of mouse specimens. D.M.W. conducted electron microscopy analysis with supervision from D.C.W. D.M.W., A.C.G., K.R.D., S.G.B., and D.M.F. interpreted all datasets. D.M.W. and A.C.G. drafted portions of the manuscript. D.M.F. conceived and designed the project, and wrote the manuscript with editorial help from D.M.W. and A.C.G.

## Competing interests

The authors declare no competing interests.

## Additional information

**Peer review information** : *Communications Biology* thanks Albert Jeltsch, Fuchun Yang and the other, anonymous, reviewer for their contribution to the peer review of this work. Primary Handling Editors: Marina Holz and Zhijuan Qiu. Peer reviewer reports are available.

