## [Peer Review File · Communications Biology]

Reviewers' comments:

Reviewer #1 (Remarks to the Author):

This is an interesting manuscript that however needs improvement.
please respond to the following points:

- 1)line 55: claiming the negative correlation is SPECIFIC is a bit of an overstatement, that the authors should contextualize.
- 2)the word rampant should be replaced all over the text, as it is used too frequently
- 3)line 97: explain why the H2009 line was adopted, what was the rationale for not choosing other lines as well. In addition explain in a concise way why H3K36me3 is of relevance.
- 4) line 119: besides YFP other proteins must be addressed to prove their statement
- 5) line 136: explain in a logic manner the statements related to lysosome and introduce the relevance of this analysis
- 6) pH3 comes out of the blue in the figures, please describe it in the result section and discuss accordingly
- 7) Please include in a Table the genes identified by pathway analyses in Figure 1a-b.
- 8) Please include real facts data showing experimental and control samples in Figure 2 besides the FACS strategy
- 9) While the point is somehow well made, efficiency of KD reported in Supplementary Figure 3 is not appropriate.

Reviewer #2 (Remarks to the Author):

The authors studied the effect of SETD2 KO in tumor cells. They provided evidence for upregulation of ribosomal and mitochondrial genes and morphological changes of the mitochondria and found higher oxidative function. Moreover, an overall enhancement of protein biosynthesis and mTORC1 signaling was observed. Based on this the authors tested mTORC1 inhibitor treatment and found tumor suppressive effects in SETD2 deficient tumors. While the data look convincing in most parts, several adjustments of the manuscript are necessary prior to a decision on publication.

Major comments:

- 1) In my assessment the writing of the manuscript is insufficient (points 1-5). The introduction is too short and does not provide the reader with an entry into the known function and activities of SETD2 and its known roles in cancer.
- 2) The results section generally is OK, but it needs subheading for structuring.
- 3) The discussion section is insufficient to put the results into context with literature data, mention limitations of the conclusions and develop some application visions.
- 4) The beginning of the results section needs to be presented in the introduction and expanded. Which tumors were analyzed? How was it ensured that matching sets of SETD2 up and down tumors are compared?
- 5) Lines 48-70: It needs to be made clear if and how the new data really extend the existing ones and which novel conclusions can be drawn on the new data. The close connection of this paper with older data impacts on novelty of the reported results.
Data description, presentation and availability must be improved.
- 6) I did not find any information on data availability and data sharing. Lack of reviewer access on data did not allow me to reproduce any of the claims.
- 7) The expression data need to be introduced with identification of differentially regulated genes, volcano plot and some key parameters regarding reproducibility.
- 8) It would be relevant to show the data in Fig. 2 and 3 also separated by animals, together with corresponding p-values. This would provide a better view on reproducibility and significance.
- 9) In Suppl. Fig. 3, SETD2 expression data (RNA and protein) should be shown for the cells treated

with the shRNA.

10) It should be clearly indicated in the text, that no survival experiments were conducted and the efficacy of the treatments was solely based on tumor area and cell counts.

Reviewer #3 (Remarks to the Author):

This manuscript investigated the mechanism in which how Setd2 inactivation in KRAS-driven lung adenocarcinoma promotes tumor cells proliferation and tumor growth. The bioinformatics analysis from published data and KRAS-driven lung tumors, the authors showed the negative correlation between Setd2 expression and oxidative metabolism and protein synthesis gene expression. They further verified the role of Setd2 deficiency in KRAS-driven lung adenocarcinoma in regulating mitochondria respiration and protein synthesis. Further, the vulnerabilities of oxidative respiration and mTORC1 were utilized for treating Setd2 deficient lung adenocarcinoma with their inhibitors.

The study is interesting for exploring how Setd2 as a tumor suppressor regulates KRAS-driven lung adenocarcinoma and applying Setd2 deficiency or mutation as a functional marker of sensitivity to inhibition of oxidative respiration and mTOR. Specific comments for this manuscript are listed below.

Major comments:

1. The authors used different tumor models in figure 2 and 3 (KP tumors in Fig.2, while K tumors in Fig.3) to demonstrate Setd2 roles in mitochondria metabolism and protein synthesis respectively. I wonder the reason for choosing different model in Fig. 2 and 3, and if similar results were acquired in there two tumor models (K and KP) for Setd2 inactivation in regulating mitochondria respiration and protein synthesis.
2. I wonder if GSEA result showed enrichment of mTOR signaling-related genes in Setd2 low expression cancers or Setd2 KO tumor cells in Fig. 1 and Supplementary Fig.1., as mTORC1 links protein synthesis and OXPHOS.
3. Setd2 expression should be blotted in Supplementary Fig. 3a to show Setd2 knockdown efficiency though H3K36me3 is the downstream target of Setd2, due to possible off-target of shRNAs.
4. In Figure 3, I suggest that western blot for 4E-BP1 phosphorylation in K-Ctrl and K-Setd2 KO tumor cells can be added to clearly show mTORC1 activity in tumor cells which can avoid other stroma cells' signal effects in tumor tissues.
5. In Figure 4b and 4e, it looks that Setd2 KO tumors treated with these inhibitors are similar or bigger than Ctrl tumors with the same inhibitors, I suggest that the authors give the p values between Ctrl and Setd2 tumors in groups with each inhibitor treatment. If no significant difference, discuss the limitation of these inhibitors for treatment of Setd2 deficiency in KRAS-driven lung adenocarcinoma.
6. Since Setd2 is H3K36me3 methyltransferase, I am interested if oxidative respiration and mTORC1 activity negatively regulated by Setd2 are dependent on its histone methyltransferase activity.

Minor comments:

1. In Fig. 1c right panel and its corresponding legend, KPY-Setd2 KO vs KPY-Ctrl was shown, but the text described KP-Setd2 KO and KP-Ctrl tumors. Please check what tumors were analyzed here.
2. In Line 76-79 of text "Although the overall between genotypes, the mitochondria by several parameters (Supplementary Fig. 2A,B).", I suggest putting "(Supplementary Fig. 2A,B)" after "between genotypes", to avoid confused reading.
3. In Fig.3a legend, Setd2KO was wrongly marked.
4. Please give the full name of Setd2 when it initially appeared in text.

ALL REFEREES: We thank all of the reviewers for their interest in our study and their prudent suggestions on how to improve our work. Most of the reviewer suggestions centered on the writing of the manuscript and identified areas that lacked important details or required expansion for reader comprehension. Further, the reviewers identified areas where data presentation could be improved to aid in the interpretation of our study. We have made all the requested changes to the manuscript and figures and as a result our paper is clearer and more easily interpretable.

The specific areas of concern indicated by each referee are listed below followed by a detailed response for each:

REFEREE 1

Q1: “Claiming the negative correlation is SPECIFIC is a bit of an overstatement that the authors should contextualize.”

A1: The negative correlation of ribosomal- and mitochondrial-associated gene sets with SETD2 status is robust; it is observed in human and mouse lung adenocarcinomas, and across multiple other major tumor types where SETD2 is frequently mutated. However, the reviewer is correct that correlations are not causations and therefore cannot be specific. We have edited the text to clarify that this observation is “associated with” SETD2 expression and that we are simply highlighting the difference between SETD2 and other common tumor suppressors in lung adenocarcinoma (Rb and p53).

Q2: “The word rampant should be replaced all over the text as it is used too frequently”

A2: We agree with the reviewer that the word “rampant” is used too frequently and have replaced it in the text.

Q3: “Explain why the H2009 line was adopted, what was the rationale for not choosing other lines as well. In addition explain in a concise way why H3K36me3 is of relevance”

A3: The human lung adenocarcinoma cell line H2009 was used in our study because it harbors an oncogenic KRAS mutation and a P53 loss-of-function mutation which is analogous to our mouse system. Further, in our experience, SETD2 inactivation is not well tolerated in many human lung adenocarcinoma lines. We have clarified our decision to use this cell line in the text.

Q4: “Besides YFP other proteins must be addressed to prove their statement [that loss of SETD2 causes increased translation]”

A4: We demonstrate that global protein synthesis is occurring at a higher rate in SETD2 deficient tumors in the mouse. The difference between control and SETD2 KO tumors is stark and the data are highly significant. The use of OP-Puro to track newly translated proteins is a gold standard approach to support our conclusion of increased protein translation. We presented additional data that makes use of an endogenous YFP reporter that is intrinsic in our KPY model. The YFP knock in allele is attractive because it is strongly expressed, easily detectable, and not expected to be impacted by position effects or genomic variations that are intrinsic to our mixed background mice. Therefore, while the OP-Puro data report on the synthesis of all proteins in the proteome which are expressed at considerable variable relative levels, the Rosa26YFP expression data report on one highly controlled reporter gene; therefore each complements the other.

Q5: “Explain in a logical manner the statements related to the lysosome and introduce the relevance of this analysis”

A5: We thank the reviewer for identifying a passage in the text that needs further context. We have edited the text to clarify the significance of looking at the localization of mTORC1 to the lysosome. In brief, it is well-established that active mTORC1 complex signals from this location.

Q6: “pH3 comes out of the blue in the figures, please describe it in the result section and discuss accordingly”

A6: We have edited the text to clarify the reason for performing this analysis. We explain the importance of p-H3 and how it marks actively proliferating cells.

Q7: “Please include in a Table the genes identified by the pathway analyses in Figure 1a-b”

A7: We have now provided a supplementary file that lists all of the differentially expressed genes in SETD2^{KO} and control tumors.

Q8: “Please include real FACS data showing experimental and control samples in Figure 2”

A8: In addition to the gating strategy we provided in Supplementary Figure 2, we have now added histograms of representative SETD2^{KO} and control tumors to the flow data presented in Figures 2 and Supplementary Figure 2.

Q9: “While the point is somehow well made, efficiency of KD reported in Supplementary Figure 3 is not appropriate”

A9: We believe that showing the efficiency of the knockdown of H3K36me3 in the SETD2 shRNA cell lines used in the OCR analysis is important to show readers. As other reviewers have suggested we have clarified the importance of H3K36me3 (the methylation mark that SETD2 deposits) in the introduction and have added new figure panels (Supplementary Figure 3A-B,E) to show the knockdown of SETD2 expression in the shRNA lines (both mRNA and protein abundance).

REFEREE 2

Q1: “In my assessment the writing of the manuscript is insufficient (points 1-5). The introduction is too short and does not provide the reader with an entry into the known function and activities of SETD2 and its known roles in cancer.”

A1: We thank the reviewer for their helpful suggestions on how to make our paper better and more readable. We have now included additional text in the introduction that details the known functions of SETD2.

Q2: “The results section generally is OK, but it needs subheading for structuring.”

A2: We thank the reviewer for this excellent suggestion to make our work more digestible. We have now broken up the results section into distinct parts with subheadings.

Q3: “The discussion section is insufficient to put the results into context with literature data, mention limitations of the conclusions and develop some application visions.”

A3: We have now added discussion of potential therapeutic implications of our work and have connected our study to the growing connection between enzymes that regulate histone post translational modifications and cancer metabolism and well as interesting gene-gene interactions that exist in human lung adenocarcinoma and mouse lung adenocarcinoma models.

Q4: “The beginning of the results section needs to be presented in the introduction and expanded. Which tumors were analyzed? How was it ensured that matching sets of SETD2 up and down tumors are compared?”

A4: Thank you for pointing out this area of confusion. We have rewritten the beginning of the results and added pertinent information to the introduction to better describe the experimental scheme and methods. Briefly, tumors of defined genotype were isolated by dissection from the lung, divided into roughly equal portions and then one portion was assessed by histological analysis and the other processed for RNA-sequencing. In this way we selected tumor stage-matched samples. It should be noted that this pipeline was in the end not necessary because within each model (K and KP) the SETD2 status did not affect the grade of the largest tumors which we focused on to ensure sufficient material for downstream analysis.

Q5: “It needs to be made clear if and how the new data really extent the existing ones and which novel conclusions can be drawn on the new data. The close connection of this paper with older data impacts on novelty of the reported results.”

A5: While the RNA sequencing data for the K model was originally part of our 2017 Cancer Research Paper, its analysis then was limited only to a comparison of SETD2 deficient tumors and tumors that were deficient for ARID1A, another chromatin modifying enzyme that is mutated in human lung adenocarcinoma. We presented only a cursory gene expression analysis using Ingenuity Pathway Analysis (IPA) to determine ontology terms that were distinct between SETD2 KO tumors and ARID1A KO tumors. Here we use Gene Set Enrichment Analysis looking for pathways that are enriched in SETD2 KO in both the K and KP mouse models as well as human tumors (TCGA). The results using GSEA are significantly more lucid in that the overlap between the mouse and human data are surprisingly similar. Looking at the data in this way, we were able to identify two pathways that are targetable vulnerabilities. These pathways were not identified previously due to the significant biases that are present in the IPA analysis and that fact that we did not use KP or TCGA data.

Q6: “I did not find any information on data availability and data sharing. Lack of reviewer access on data did not allow me to reproduce any of the claims.”

A6: We have included a supplemental file that contains all of the differentially expressed genes in SETD2^{KO} and control tumors for both the K and KP tumor models. Raw data will be made available in the Gene Expression Omnibus for public scrutiny after acceptance of the manuscript per the publisher and NIH guidelines.

Q7: “The expression data need to be introduced with identification of differentially regulated genes, volcano plot and some key parameters regarding reproducibility.”

A7: We have expanded this section of the manuscript to better describe the analysis and data interpretation. We have added volcano plots to support Figure 1 in the supplementary material.

We have also included a supplemental file that highlights differentially expressed genes in the K and KP models.

Q8: “It would be relevant to show the data in Fig. 2 and 3 also separated by animals, together with corresponding p-values. This would provide a better view on reproducibility and significance.”

A8: We chose not to stratify the data presented in Figures 2 and 3 by mouse because we found no differences between animals within experimental groups. For the reviewer we have provided the data as requested here (See **Rebuttal Figure 1**). We hope the reviewer agrees that this strategy is more confusing for the average reader and that for clarity we should keep the data grouped by genotype or condition.

Q9: “In Suppl. Fig. 3, SETD2 expression data (RNA and protein) should be shown for the cells treated with the shRNA.”

A9: We have updated Supplemental Figure S3 to include SETD2 RNA and protein abundance for the SETD2 shRNA cell lines.

Q10: “It should be clearly indicated in the text, that no survival experiments were conducted and the efficacy of the treatments was solely based on tumor area and cell counts.”

A10: We have added this statement to the manuscript.

REFEREE 3

Major comments:

Q1: “The authors used different tumor models in figure 2 and 3 (KP tumors in Fig.2, while K tumors in Fig.3) to demonstrate Setd2 roles in mitochondria metabolism and protein synthesis respectively. I wonder the reason for choosing different model in Fig. 2 and 3, and if similar results were acquired in there two tumor models (K and KP) for Setd2 inactivation in regulating mitochondria respiration and protein synthesis.”

A1: In our analyses in Figures 2 and 3, we were limited by the small size of *K* tumors with wildtype SETD2. We reasoned that the small size of these tumors would make it difficult to easily perform the analyses in Figure 2. By using the KP model, we were able to capitalize on their larger size to facilitate analysis. The spirit of the concern is well taken however, because if differences existed between the models it would offer insightful into the biology of SETD2 in lung adenocarcinoma. Importantly, we found in the GSEA pathway analyses presented in Figure 1 that both *K* and *KP* tumors with inactive SETD2 have an enrichment of genes associated with OXPHOS and protein synthesis. This suggests that SETD2 inactivation in both the *K* and *KP* models has similar effects on mitochondrial metabolism, OXPHOS and protein synthesis.

Q2: “I wonder if GSEA result showed enrichment of mTOR signaling-related genes in Setd2 low expression cancers or Setd2 KO tumor cells in Fig. 1 and Supplementary Fig.1., as mTORC1 links protein synthesis and OXPHOS.”

A2: We thank the reviewer for this suggestion. In fact, focusing on the investigator curated gene sets that are part of MSIGdb and GSEA analysis, we have found that mTORC1 signaling-related genes are highly enriched in SETD2-deficient tumors. We have added this insight into the manuscript in Supplementary Figure S4E.

Q3: “Setd2 expression should be blotted in Supplementary Fig. 3a to show Setd2 knockdown efficiency though H3K36me3 is the downstream target of Setd2, due to possible off-target of shRNAs.”

A3: We have updated Supplemental Figure 3 to include SETD2 RNA and protein abundance for the SETD2 shRNA cell lines.

Q4: “In Figure 3, I suggest that western blot for 4E-BP1 phosphorylation in K-Ctrl and K-Setd2 KO tumor cells can be added to clearly show mTORC1 activity in tumor cells which can avoid other stroma cells’ signal effects in tumor tissues.”

A4: We appreciate the reviewer’s suggestion because it is important to distinguish cancer cell from stromal cell phenotypes. Our approach was to use immunostaining to distinguish tumor cells from stromal cells as this is an analysis that can detect distinct cell types. Western blotting of tumor tissue is highly problematic as it is an ensemble measurement of proteins that are found in cancer cells and in stromal cells. This strategy would have the opposite effect of what was intended by the reviewers question, *i.e.* we would be measuring both tumor AND stromal p-4E-BP1 which would lead to less interpretable results.

Q5: “In Figure 4b and 4e, it looks that Setd2 KO tumors treated with these inhibitors are similar or bigger than Ctrl tumors with the same inhibitors, I suggest that the authors give the p values between Ctrl and Setd2 tumors in groups with each inhibitor treatment. If no significant difference, discuss the limitation of these inhibitors for treatment of Setd2 deficiency in KRAS-driven lung adenocarcinoma.”

A5: We apologize for the confusion. The important point here is that IACS-10759, rapamycin, and metformin each suppress *K-Setd2*^{KO} tumor growth. The suppression of tumor growth is significant compared to untreated *K-Setd2*^{KO} tumors and the statistics with p value are indicated in the figure. We believe the point of confusion comes from the fact that none of these treatments limited *K-Ctrl* tumor growth. As such, it is perhaps not surprising that the drugs do not reduce *K-Setd2*^{KO} tumor growth to levels equivalent to *K-Ctrl*. The data indicate that SETD2 loss specifically promotes OXPHOS and mTORC1 signaling and that these are responsible for promoting tumor growth. Thus when inhibited, we are blocking the effects of SETD2 inactivation.

Q6: “Since Setd2 is H3K36me3 methyltransferase, I am interested if oxidative respiration and mTORC1 activity negatively regulated by Setd2 are dependent on its histone methyltransferase activity.”

A6: The fact that SETD2 is not only a histone methyltransferase but also a methylase for transcription factors, cytoskeletal proteins, and other chromatin modifying enzymes makes this an extremely difficult question to answer. This is a major research project in the lab but we believe the answer is complex— *i.e.* some features of tumor suppression rely on histone methylation and others on non-histone targets. When one looks specifically at human cancer derived mutations, it is clear that many cluster in the catalytic SET domain of SETD2 which would suggest the methyltransferase activity is key to its tumor suppression. More research way beyond the scope of this paper is needed to answer this very interesting question.

Minor comments:

Q1: “In Fig. 1c right panel and its corresponding legend, KPY-Setd2 KO vs KPY-Ctrl was shown, but the text described KP-Setd2 KO and KP-Ctrl tumors. Please check what tumors were analyzed here.”

Q2: “In Line 76-79 of text “Although the overall between genotypes, the mitochondria by several parameters (Supplementary Fig. 2A,B).”, I suggest putting “(Supplementary Fig. 2A,B)” after “between genotypes”, to avoid confused reading.”

Q3: “In Fig.3a legend, Setd2KO was wrongly marked.”

Q4: “Please give the full name of Setd2 when it initially appeared in text.”

A1-4: We have made all of the minor corrections to the manuscript and figures and thank the reviewer for their thorough review of our data.

Rebuttal Figure 1

(Figure 2b-g)

(Supplementary Figure b,d-e)

(Figure 3b,d,f)

Reviewers' comments:

Reviewer #1 (Remarks to the Author):

The manuscript has significantly improved and it is an interesting one. However I would have appreciated having Figure 2a and Figure 3a generated from both murine models. I highly encourage including those data in the current manuscript.

Also, but not necessary in this manuscript, Western Blot in Figure 3 could have been performed on Epcam+ cells, in order to simply study the cancer versus the stroma component. It is something the authors should keep in mind for future studies.

Last, the HE Figure 3 for IACS does not appear to be a representative one.

Minor: some minor English polishing still needs to be done.

Reviewer #2 (Remarks to the Author):

This is an interesting manuscript reporting important data. Most of my comments were sufficiently addressed and I recommend acceptance of this work. However, I still feel it is not good practice, not providing a confidential reviewer access to the primary data.

Reviewer #3 (Remarks to the Author):

Minor comments:

Q1: "In Fig. 1c right panel and its corresponding legend, KPY-Setd2 KO vs KPY-Ctrl was shown, but the text described KP-Setd2 KO and KP-Ctrl tumors. Please check what tumors were analyzed here." Current text: "K-Setd2KO tumors, which were low grade adenomas, and KPY-Setd2KO tumors, which were higher grade adenocarcinomas, had vastly different gene expression profiles than their K-Ctrl and KP-Ctrl counterparts and a strong enrichment of ribosomal- and mitochondrial-associated gene sets. (Fig. 1B-C, Supplementary Fig. S1H-I)."

"KP-Ctrl" should be "KPY-Ctrl" if "KPY-Setd2KO" is correct.

No more concerns.